# A Regional-Scale Index for Assessing the Exposure of Drinking-Water Sources to Wildfires

**François-Nicolas Robinne** [1,*] , **Kevin D. Bladon** [2] , **Uldis Silins** [3], **Monica B. Emelko** [4] ,
**Mike D. Flannigan** [5] , **Marc-André Parisien** [6], **Xianli Wang** [7], **Stefan W. Kienzle** [8,9] and
**Diane P. Dupont** [10]

1   Canada Wildfire, Renewable Resources, 751 General Services Building, University of Alberta, Edmonton,
    AB T6G 2H1, Canada
2   Department of Forest Engineering, Resources, and Management, Oregon State University, Corvallis,
    OR 97331, USA; bladonk@oregonstate.edu
3   Department of Renewable Resources, University of Alberta, Edmonton, AB T6G 2H1, Canada;
    uldis.silins@ualberta.ca
4   Department of Civil and Environmental Engineering, University of Waterloo, Waterloo, ON N2L 3G1,
    Canada; mbemelko@uwaterloo.ca
5   Canada Wildfire, Department of Renewable Resources, University of Alberta, Edmonton, AB T6G 2H1,
    Canada; mike.flannigan@ualberta.ca
6   Natural Resources Canada, Canadian Forest Service, Northern Forestry Centre, 5320 122 St., Edmonton,
    AB T6H 3S5, Canada; marc-andre.parisien@canada.ca
7   Natural Resources Canada, Canadian Forest Service; Great Lake Forestry Centre, 1219 Queen Street East,
    Sault Ste. Marie, ON P6A 2E5, Canada; xianli.wang@canada.ca
8   Department of Geography, University of Lethbridge, Alberta Water and Environmental Science Building,
    4401 University Drive, Lethbridge, AB T1K-3M4, Canada; stefan.kienzle@uleth.ca
9   Applied Behavioral Ecology and Ecosystems Research Unit, University of South Africa, P.O. Box 392, Florida,
    Pretoria 1710, South Africa
10  Faculty of Social Sciences, Brock University, Niagara Region, 1812 Sir Isaac Brock Way, St. Catharines,
    ON L2S 3A1, Canada; diane.dupont@brocku.ca
*   Correspondence: robinne@ualberta.ca; Tel.: +1-587-589-6449

**Abstract:** Recent human-interface wildfires around the world have raised concerns regarding the reliability of freshwater supply flowing from severely burned watersheds. Degraded source water quality can often be expected after severe wildfire and can pose challenges to drinking water facilities by straining treatment response capacities, increasing operating costs, and jeopardizing their ability to supply consumers. Identifying source watersheds that are dangerously exposed to post-wildfire hydrologic changes is important for protecting community drinking-water supplies from contamination risks that may lead to service disruptions. This study presents a spatial index of watershed exposure to wildfires in the province of Alberta, Canada, where growing water demands coupled with increasing fire activity threaten municipal drinking-water supplies. Using a multi-criteria analysis design, we integrated information regarding provincial forest cover, fire danger, source water volume, source-water origin (i.e., forested/un-forested), and population served. We found that (1) >2/3 of the population of the province relies on drinking-water supplies originating in forested watersheds, (2) forest cover is the most important variable controlling final exposure scores, and (3) watersheds supplying small drinking water treatment plants are particularly exposed, especially in central Alberta. The index can help regional authorities prioritize the allocation of risk management resources to mitigate adverse impacts from wildfire. The flexible design of this tool readily allows its deployment at larger national and continental scales to inform broader water security frameworks.

**Keywords:** post-fire hydrology; source water protection; drinking-water security; multi-criteria analysis; "*Forests to Faucets*"; community drinking-water; compound wildfire-water risk

## 1. Introduction

Increasing rates of global environmental change have contributed to widespread water-security challenges [1]. For example, increased occurrence and severity of droughts, pronounced glacial melt, sinking deltas, extraction of timber and other natural resources from forests, wetland urbanization, depleted fisheries, and intensification of agriculture, along with increased water demand associated with population growth, have all increased the vulnerability of the world's water supply [2–4]. Unfortunately, new challenges are also emerging. Over the past two decades a growing number of extreme wildfire events have occurred in many parts of the world [5,6], which have increased concern about the magnitude and longevity of effects on water resources and aquatic ecosystem health [7]. Burned catchments may experience a range of hydrological and morphological perturbations depending on burn severity, area burned, catchment physiography, and post-fire precipitation regimes [8]. Many studies have illustrated post-fire impacts on soil hydraulic properties and runoff [9–11], sediment and nutrient concentrations [12,13], and the occurrence of floods and debris flows [14]. Such perturbations can threaten the reliability of downstream water supply for community needs [15,16], with some of these impacts lasting for a decade or more [17,18].

In most jurisdictions, the provision of safe drinking water requires disinfection at a minimum; in the case of surface water supplies, conventional treatment is typically comprised of a series of physico-chemical processes described as "chemically-assisted filtration" that are followed by disinfection [16,19]. The specific treatment process configuration for a community is determined, in part, by the current and projected availability and quality of the source (i.e., raw) water supply [16]. However, severe wildfires can lead to increasingly variable and deteriorated drinking water source quality, potentially leading to substantial challenges for drinking water treatment processes [16,20,21]. In particular, greater variability and increased concentrations of total suspended solids and turbidity, dissolved organic carbon (DOC), and other nutrients such as phosphorus (that can promote algal blooms) after a wildfire can challenge treatment process performance and increase operational costs [16,18]. For instance, treatment of lower-quality source water coming from burned areas may result in difficulties for treatment plants to meet chemical coagulant demand [16] and in greater production of disinfection by-products [21], some of which are a public health concern (if they pass into distributed water supplies) due to their alleged carcinogenic effects from prolonged exposure [22]. In the worst-case scenarios, poor source-water quality following wildfire can force treatment plants to shut down or shift to other water sources so drinking-water delivery to consumers is maintained [23,24].

As such, in regions where community drinking-water supply originates in forests, it is increasingly important to identify and evaluate the potential hazards to municipal treatment systems from post-fire water contamination. Identifying locations where there is a greater probability of adverse effects on key drinking-water sources from wildfire can facilitate coordination of forest management activities and utility operations to mitigate threats from wildfire and ensure protection and distribution of safe drinking water [25]. Moreover, this knowledge can contribute to enabling water treatment vulnerability assessments, developing strategies to rapidly identify water contamination, and responding to the unique challenges often associated with post-wildfire drinking-water treatment [26]. Such knowledge will be critical in adapting to the growing pressures from global environmental change on forest ecosystems and water resources, and consequently, on drinking-water systems [27].

Water and land managers currently lack adequate tools for assessing potential, wildfire-associated risks to municipal drinking-water systems served by forested catchments. As a result, there have been several efforts recently to use geospatial information to develop spatial indices for improving assessments of wildfire hazard and exposure of municipal watersheds to wildfire [28–30]. In 2009,

a spatial multi-criteria method was used in Eastern USA to quantify source water provision from forested watersheds and rank them according to their exposure to multiple anthropogenic stressors [31]. This approach was later modified in the 'Forests to Faucets (F2F)' initiative by the US Forest Service to highlight the importance of forests to downstream drinking-water supplies [32]. The primary objectives of F2F were to explore spatial relationships between forested water sources, the number of downstream consumers, and potential threats from urban development, insects and disease, and wildfire potential [33,34]. While this model was a valuable initial effort, its direct applicability outside of USA is constrained, mainly because of limitations in data availability—yet, the global demand for spatially explicit information combining wildfire hazard and associated risks to community water supply is growing [35].

Here, we provide a spatial index, the Source Exposure Index (*SEI*), based on a generic multi-criteria framework to assess the exposure of forested watersheds to wildfire hazard. We used the definition of exposure proposed by the United Nations as "the situation of people, infrastructure, housing, production capacities, and other tangible human assets located in hazard-prone areas" [36]. Specifically, we aggregated information pertaining to water availability, water demand, forest cover, and fire danger in watersheds that supply surface water to 94 downstream communities in Alberta, Canada. The *SEI* is conceptually similar to F2F but differs in the source of data and their integration. Our objectives were to: (a) evaluate the exposure of source water supplying communities downstream of wildfire risk, and (b) develop and propose the first module of a larger pan-Canadian wildfire-water risk assessment framework. In contrast to previous efforts, our index identifies the spatial location of drinking-water supply intakes and relates these to forested water sources and their wildfire danger history. We believe that the flexibility of the index will facilitate its coupling with other water-resource indicators, as well as its integration into a Canadian water-security framework assessing various environmental-change scenarios and their potential impacts on national freshwater resources.

## 2. Materials and Methods

### 2.1. Study Area

The province of Alberta encompasses a large forested region (approximately 2/3 of the 661,848 km$^2$ total provincial land area) in Boreal Plain and Montane Cordillera ecozones in the Northern and Western regions [37]. The climate of Alberta reflects the generally cold, dry, continental conditions characteristic of Northern interior regions East of the Canadian Rocky Mountains. Mean annual precipitation (510 mm yr$^{-1}$) varies spatially, reflecting the diverse physiography and forests across the province. The highest annual precipitation (700–1400 mm yr$^{-1}$) generally coincides with the highest elevations (3747 m above sea level maximum) within the Rocky Mountains (Figure 1). Comparatively, the lowest mean annual precipitation (325–400 mm yr$^{-1}$) tends to occur in the lower elevation regions (210 MASL minimum) in the Northeast [38]. The hydrologic regime across the entire region is generally snowmelt-dominated with the greatest stream/river flows occurring during May–June, coincident with the late snowmelt freshet and onset of early summer convective and frontal storm activity. Strong variation in mean annual water yield across the province (29–936 mm yr$^{-1}$, [39]) reflects the integrated interaction between strong spatial gradients in elevation, temperature, precipitation, topography, and geology.

Forest vegetation is characteristic of Northern Boreal and Northern temperate Rocky Mountain regions. The Northern forests of the province are mixedwoods or conifer dominated, where the main tree species in upland landscapes are *Populus tremuloides* Michx., *P. balsamifera* L., *Picea glauca* (Moench) Voss, and *Pinus banksiana* Lamb. Extensive peatlands in the North are dominated by *Picea mariana* (Mill.) B.S.P. (Pinaceae) and *Larix laricina* (Du Roi) K. Koch. Forests in the Western regions of the province are dominated by mixed conifer (*Picea engelmannii* Parry ex Engelm. and *Abies lasiocarpa* (Hook.) Nutt.) in the high elevation Rocky Mountain front-range, lodgepole pine (*Pinus contorta* Douglas ex Loudon) at

mid-elevations, and broadleaf and mixedwood forest (*Populus tremuloides*, *Pinus contorta*, *Picea glauca*) in the lower-elevation foothills.

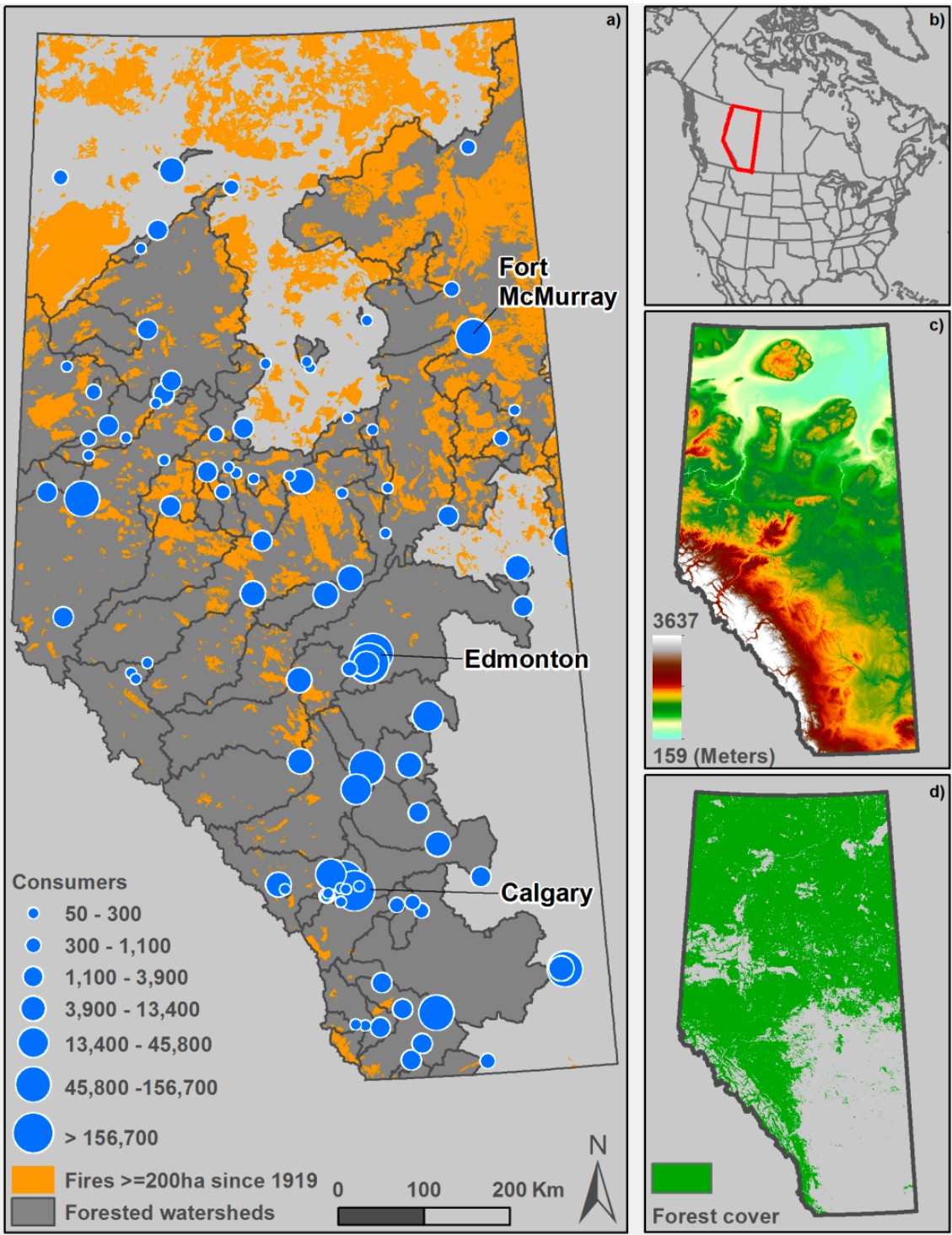

**Figure 1.** Details of the study area, including: (**a**) the location of drinking-water treatment facilities reliant on surface water supplies from forested watersheds in Alberta (symbol size indicates population size class served by the utility) and historic spatial distribution of large fires (>200 ha) [40]; (**b**) the location of Alberta in North America; (**c**) Alberta terrain (elevation); and (**d**) Alberta forest cover.

The fire regime of the province is characterized by low-frequency stand-replacing large fires that mostly occur between early April and late September [41]. Over the 2006–2015 period, the average

annual area burned was over 280,000 hectares mostly due to lightning-caused fires, although human ignitions have become increasingly influential [42–44]. Historically, large fires occurred more frequently in the Northern boreal region relative to the Southern areas of the province [45,46]. However, fire also remains the dominant natural disturbance agent in the Southwestern headwaters of the Rocky Mountain forest region [47] (Figure 1). Recent analyses of national fire records and future climate projections suggest that fire activity in Alberta is likely to increase, with longer fire seasons and a greater potential for large fires [48,49].

The majority (92%) of drinking water supply for Alberta municipalities is provided by surface waters (rivers, lakes) [50], with the majority of large municipalities water-supply originating from forested regions. However, with the exception of the Regional Municipality of Wood Buffalo (which includes the city of Fort McMurray), most of the large municipalities are located in agricultural or parkland regions of the province, which are downstream of forests. Municipal water consumption has increased steadily with population growth (currently ~4.3 million residents) in recent decades. Projected future population growth, along with limits on new water allocation, are expected to constrain municipal development in the province, particularly in the Southern region [51,52].

### 2.2. Data Preparation

#### 2.2.1. Community Water Systems

The first step in our assessment of drinking-water exposure to wildfire was to identify the source watersheds for each community in the province [53]. Using data provided by the Government of Alberta, we created an ad hoc watershed layer based on the geolocation of the intake for each municipal drinking-water treatment plant reliant on surface water sources (Figure 1). We excluded utilities reliant upon groundwater (including shallow groundwater) and groundwater under the direct influence of surface water sources from this analysis because of the lack of information regarding wildfire impacts on subsurface water supplies. In total, we identified 124 drinking water utilities using surface water sources in the province.

The contributing area of each source watershed was delineated using ArcHydro for ArcGIS 10.X [54,55]. Given the focus of the analysis, we intersected each source watershed with a forest cover layer and retained only the water utilities served entirely or partially by forested sources, which reduced the number of utilities to 94 (Figure 1). Hereafter, we refer to our study catchments as watersheds of interest (WOI). We then used data from the 2006 and 2011 Census of Canada to determine the total population served by each drinking water utility, as well as the population served in each region (Table 1).

**Table 1.** Datasets used in the Source Exposure Index (*SEI*). Mean and Standard Deviation (SD) are for the raster grids extracted using the forested watersheds only.

| Variable | Proxy | Source | Year | Unit | Mean (SD) |
|---|---|---|---|---|---|
| Consumers | Source watershed & watershed importance | Government of Alberta, Government of Canada | 2013 | Number of people | 26,000 (94,900) |
| Watersheds | Distance to water intake | Government of Alberta | 2013 | km$^2$ | 15,000 (27,000) |
| Water yield | Quantity of water supply provided per watershed | Environment Canada, University of Lethbridge | 2013 | m$^3$ km$^{-2}$ yr$^{-1}$ | 116,000 (125,000) |
| Forest percent cover | Protective forest cover per watershed | Alberta Biodiversity Monitoring Institute, Environment Canada | 2013 | % | 52.2 (33.3) |
| Fire Weather Index | Extreme fire hazard threatening water protection forests | Environment Canada, Canadian Forest Service | 2015 | Unitless | 26.7 (9.2) |

### 2.2.2. Water Yield

The annual water yield, expressed in 1000 m$^3$ km$^{-2}$ yr$^{-1}$ (equivalent to 1 mm yr$^{-1}$), was derived from a dataset by Kienzle and Mueller [56]. Daily streamflow records for Alberta's unregulated rivers were assembled from the Water Survey of Canada public database for the period 1971–2000 [39] (Table 1). Missing winter flows were estimated by linear interpolation of daily streamflow between the last and first day of available observations. The errors of these estimations were considered to be small relative to the mean annual streamflow. Where watersheds are regulated, naturalized streamflow time series were used, which were computed by Alberta Environment for the time period 1912–2001 [57] by in-filling data gaps and correcting streamflow due to known anthropogenic influences, such as water withdrawals, diversions, and reservoirs.

The original dataset provided streamflow data for 292 gauged sub-watersheds. Mean annual water yield was determined by dividing the mean annual streamflow volumes by the respective watershed areas [58], enabling comparisons of water production between watersheds. Streamflow volumes could only be directly related to the respective watersheds in headwater catchments with no upstream inflows. For nested watersheds with upstream gauging stations, flow volumes measured at upper gauging stations (i.e., inflows) were subtracted from the measured volumes at downstream stations and divided by the partial contributing watershed areas between stations. The resulting watershed layer containing mean annual water yield values was converted to a one km$^2$ spatial grid, whose values were then averaged for each WOI (Figure 2a). Areas showing a negative water yield, where water inputs to the watersheds were greater than the outflow, were given a 0.1 value to avoid negative scores in the final index.

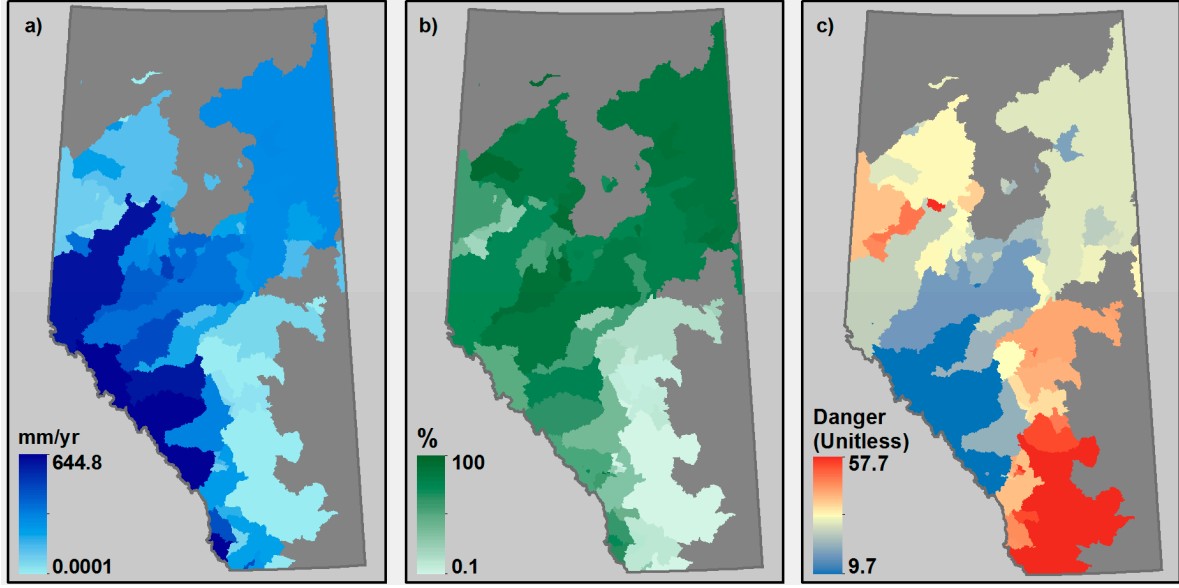

**Figure 2.** Grid inputs used in the Source Exposure Index, including: (**a**) the water yield per unit area, (**b**) the percent forest cover per watershed, and (**c**) the mean of the 95th percentile of the fire danger per watershed of interest. We only used raster grid values located in the watersheds of interest in the calculation of the Source Exposure Index.

### 2.2.3. Forest Cover

Forest cover information was extracted from the Alberta Biodiversity Monitoring Institute (ABMI) Wall-to-wall 2010 land cover vector map, which was derived from Landsat 30-m satellite imagery and developed for regional-scale environmental assessments [59]. Based on the 11 land-cover classes available in the dataset, we selected the classes providing information on coniferous, broadleaf, and mixed forest types. We also included the shrubland cover class, as it often represents young or

low-canopy forests and post-fire forest regeneration [59]. The polygons pertaining to those classes were then dissolved into a single-value forest cover layer and converted to a raster grid. We compared the ABMI land cover layer with large fire (≥200 ha) perimeters from the National Fire Database polygon layer between 1980–2016 [40], as it is common in many land-cover products to find burned areas classified as non-forested, especially in recently burned regions. Gaps (i.e., no data) in the forest cover where large fire perimeters overlapped were coded as forest as a simple way to account for post-fire vegetation recovery. We used this forest cover layer to calculate the percent forest cover for each WOI (Figure 2b).

### 2.2.4. Fire Danger

The fire danger data used in this study consisted of raster grids of the Fire Weather Index (FWI) System components. This system is one of two major components of the Canadian Fire Danger Rating System (CFFDRS), the other being the Fire Behavior Prediction (FBP) System (not considered in this study). The FWI System's components are calculated from daily weather conditions (temperature, relative humidity, wind speed, and 24-h precipitation); these may, in turn, be used in conjunction with data representing flammable vegetation (i.e., fuels) and topography by the FBP System to calculate quantitative measures of fire behavior (e.g., rate of spread, fire intensity). The FWI System is composed of three fuel moisture codes and three fire behavior indices [60]. The three codes, the Fine Fuel Moisture Code (FFMC), the Duff Moisture Code (DMC), and Drought Code (DC) represent the fuel moisture of surface, intermediate, and deep soil layers, respectively. The Initial Spread Index (ISI) is a wind-based indicator of fire danger, whereas the Buildup Index (BUI) is chiefly drought based. The Fire Weather Index (FWI) is an integrated indicator of overall fire danger computed from the ISI and BUI. The Canadian fire-weather database, an interpolated raster product of daily fire weather at a 3-km resolution, was provided by the Canadian Forest Service from historical data, based on surface (i.e., weather station) observations between April 1 and September 30 from 1981 to 2010 [61]. The gridded FWI System components were calculated from the gridded weather data using the fwiRaster function from the "cffdrs" R package [62], which was developed to calculate the components of the Canadian Forest Fire Danger Rating System [61].

Although the FWI System components are calculated solely from meteorological information, they are linked to several facets of fire activity and fire behavior in Canada [63]. For instance, the FFMC, which is sensitive to short-term (i.e., sub-daily) moisture fluctuations, is a strong predictor of fire ignitions [64]. The DC, being an index of drought, is strongly related to monthly area burned [65]. The DMC has been used as an indicator of fire extinguishment [66]. The FWI, as an overall index of fire danger (i.e., not to be confused with the FWI System), is a good predictor of fire activity, but has also been used as a proxy to fire intensity, a measure of energy release that is, in turn, related to the ecological impacts of fire and to biomass loss (i.e., fire severity) [67]. The FWI System thus provides a suite of meaningful and easy-to-compute proxies to fire hazard.

We used the 95th percentile to capture the relative frequency of days conducive to high or extreme wildfire behavior. Wildfires are often driven by extreme conditions and it is during those days of particularly hot, dry, and windy conditions (captured by the FWI) that most of the area burns in the boreal forest [68,69]. The 95th percentile of the FWI was calculated for each grid cell from the ensemble of daily grids, representing extreme conditions that have been encountered less than 5% of the time between 1981 and 2010. Grid cell values were then averaged for each WOI (Figure 2c).

### 2.3. Data aggregation

The Source Exposure Index (*SEI*) is a multi-criteria (i.e., composite) spatial index based on an incremental, multi-step data aggregation process. In other words, the output of each step is the product of the output of the previous step and the information provided from one of the gridded variables (e.g., water yield). We refer to the result (product) of each step as an interim indicator until the final index is produced (Figure 3).

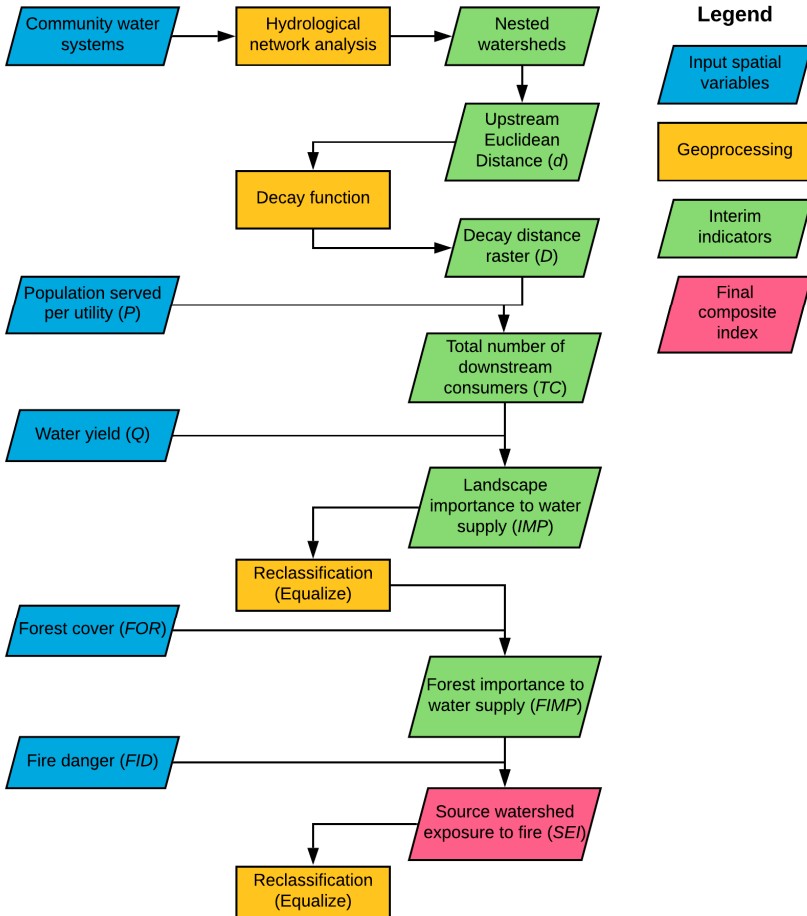

**Figure 3.** Flowchart describing the incremental steps for the creation of the Source Exposure Index (*SEI*), representing inputs, geoprocessing, and outputs. Adapted from Weidner and Todd (2011). Note that interim indicators are created following an incremental design that builds on the output of the previous step and updates it with the values of a new input. Blue parallelograms represent additional inputs, rectangles represent processes, green parallelograms represent interim indicators, and pink parallelogram represents the final composite index.

The *SEI* is primarily based on a simplified representation of the dilution processes that occur along a river network due to the contribution of additional water from tributaries. This means that the further upstream the disturbance occurs, the less likely the downstream water quality will be affected. In other words, we considered contaminant concentration to be an inverse function of the distance to the emission source. Dilution processes were represented for each WOI by first computing the Euclidean distance from the water intake (i.e., the outlet) and then transforming the distance using a decay function, so that locations directly upstream of the intake are given the most influence [34]:

$$D_n = 0.99^{(d)} \tag{1}$$

where $D_n$ was the resulting decayed distance raster grid for a *n*th WOI and *d* was the Euclidean distance raster grid values for that WOI.

We then added population (i.e., consumers) information to calculate the interim indicator *TC*, which was the total number of water consumers within a given WOI. As water yield in a given watershed may also be used in a downstream watershed, we had to account for the fraction of downstream consumers wholly or partially reliant on upstream sources, as those water supplies could also be affected by contamination from wildfire in the upstream catchment. The total number of

consumers supplied by a given WOI was calculated as the sum of the people in the WOI and the distance-weighted fraction of the consumers in downstream watersheds, computed as follow:

$$TC = \sum_{n=1}^{N} (D_n \times P_n) \tag{2}$$

where $P_n$ was the population served within the $n$th downstream watershed $D_n$ [34].

We then included water yield data ($Q$) to account for surface-water availability and calculated the interim indicator $IMP$, which was a measure of upstream landscape importance to the provision of surface drinking water to downstream consumers, as follows:

$$IMP_n = (Q_n) \times TC_n \tag{3}$$

where $Q_n$ was the water available per unit area for the $n$th nested watershed. $IMP$ provides information on the distribution of the consumers in the region and the location of major source watersheds. The result was scaled to 0–100 values using an equalization stretch.

We then calculated the interim indicator $FIMP$, which was the importance of upstream forested landscape to drinking-water supply using the following equation:

$$FIMP_n = \frac{(IMP_n) \times (FOR_n)}{100} \tag{4}$$

where $FOR$ was the percent forest cover per watershed. The output was a 0–100 grid layer showing the importance of forests to the drinking-water supply.

Finally, we calculated the Source Exposure Index ($SEI$), which provides the overall integrated index of the exposure of forested watersheds providing source water to downstream communities to wildfire using the following equation:

$$SEI = \frac{(FIMP_n \times FID_n)}{100} \tag{5}$$

where $FID$ was the fire danger for a given watershed as provided by the FWI and reclassified to a 0–100 scale prior to inclusion in the calculation. We rescaled $SEI$ to a 0–100 range using an equalization stretch. The scaled $SEI$ represented the final index, which accounts for the number of drinking-water consumers in and downstream of each catchment, the dependence of downstream catchments on upstream sources of water, the surface water resources available, the forested areas contributing to the supply of this water, and the fire danger in each forested catchment.

We also performed a simple sensitivity analysis (SA) of the $SEI$ to assess the changes in output values associated with a controlled alteration of the input values [70]. Although SA can take many different forms, we focused on ranking (i.e., importance) the four variables (i.e., population, water yield, forest cover, and fire danger) to understand their influence on the spatial pattern of the index. First, we used the "Band Collection Statistics" tool in ArcGIS to quantify the correlation between the final index and the variables. Based on these results, variables with a correlation coefficient >0.5 were assigned a constant value—in this case, their mean value—and the index was computed again, one constant variable at a time to decipher the influence of remaining variables on the final pattern of the index.

## 3. Results

Forested watersheds are critically important for downstream water supply in the province of Alberta, with 94 surface water utilities relying entirely or partially on water from forested regions. Approximately 70% of these utilities serve smaller communities (<5000 people), while 30% serve a large proportion of the Alberta population in major urban centers and through regionalization of

drinking water supplies to nearby communities. In total, these 94 waterworks systems serve >2/3 or ~2.4 million of Alberta's ~3.6 million people (2011 census). Moreover, on a flow/population weighted basis, these data indicate that 50% of people in Alberta are completely reliant on drinking water from forested sources.

Larger watersheds tended to have lower exposure values; a fact likely due to the greater influence of dilution processes when watershed size increases (Figure 4a). Higher mean *SEI* scores (i.e., ≥50 and up to 99) occurred in source watersheds mostly located in the Rocky Mountains in Southern Alberta, where watersheds tend to have a forest cover of ~60%, a fire danger >28, and water yields >180 mm/year. Those highly-exposed watersheds supply 12 small (<5000 consumer) to very small (<500 consumers) communities. Fifty-two watersheds have a mean *SEI* score ranging from 10 to 50, with a forest cover ~61%, a fire danger >25, and water yields around ~100 mm/year. Seventy-seven percent of these watersheds primarily serve small to very small communities across the north-central region of the province. Finally, there are 30 watersheds with mean *SEI* values lower than 10, which are located in the south-central and the south-east regions of the province, around the city of Edmonton and east of the city of Calgary. Those watersheds have a forest cover ~34% on average, a fire danger >26, water yields >115 mm/year, and 64% serve small to very-small communities. Overall, approximately 48% of source watersheds that supply water to large communities (>5000 people) have *SEI* scores lower than 10 (Figure 5).

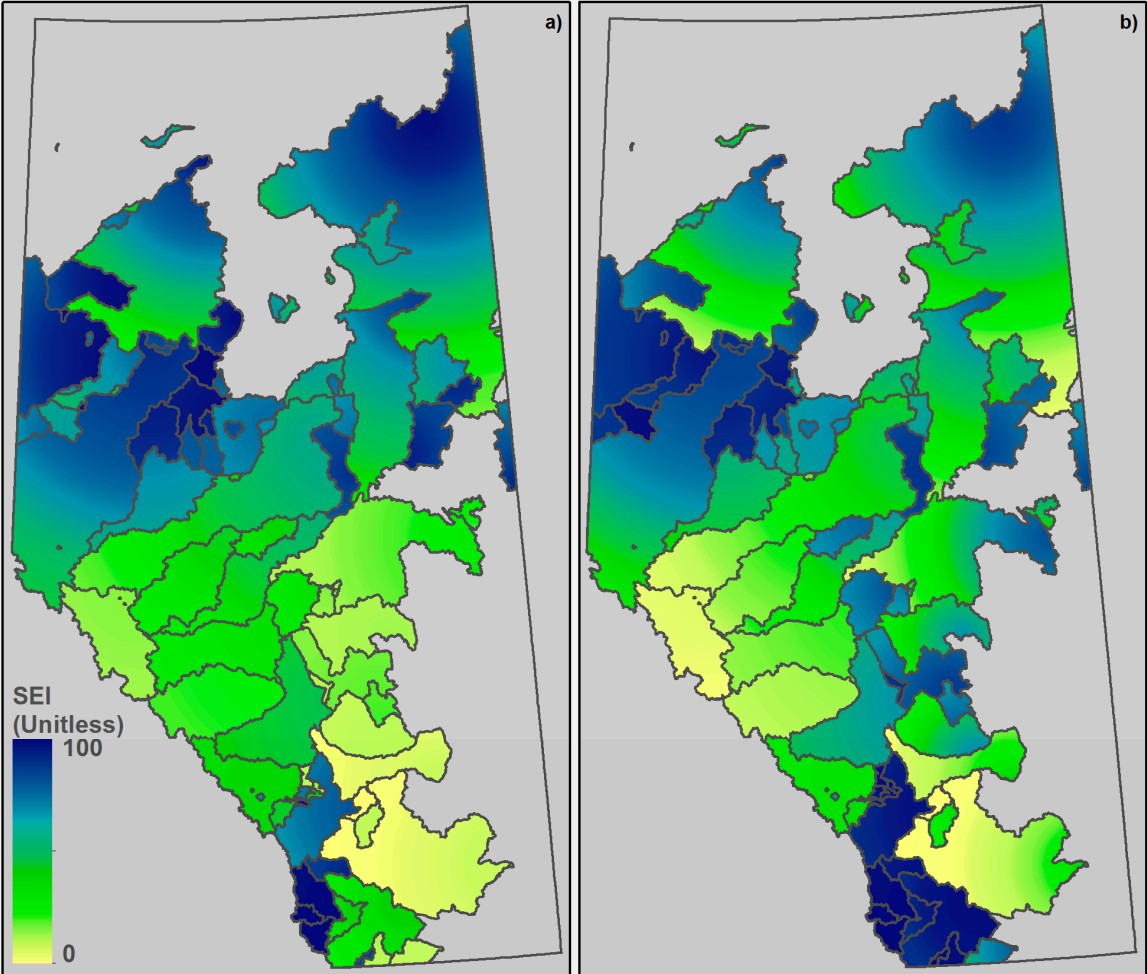

**Figure 4.** (**a**) Source Exposure Index for forested watersheds providing drinking water in Alberta. Higher values indicate higher exposure levels. (**b**) Wildfire exposure index for forested watersheds providing drinking water in Alberta as a result of the sensitivity analysis where the forest layer was set to its mean value. Higher values indicate higher exposure levels.

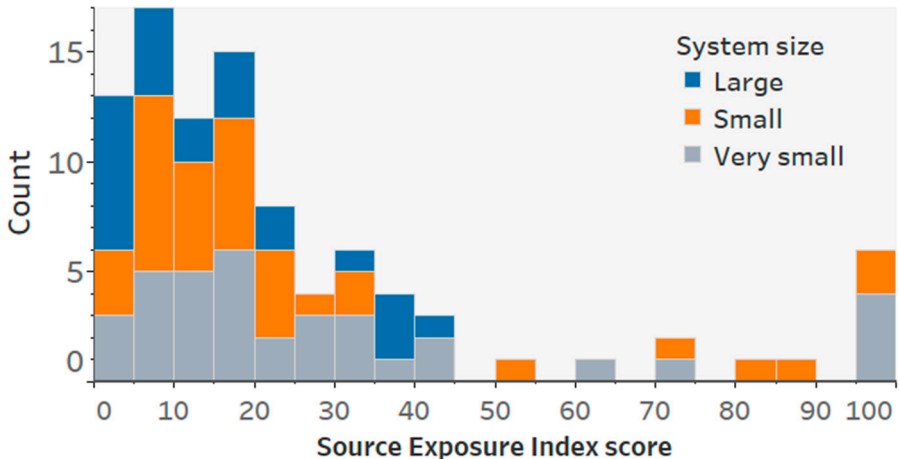

**Figure 5.** Distribution of Source Exposure Index scores for utilities serving variable number of consumers, with very small, small, and large systems serving less than 500, 500 to 5000, and more than 5000 people, respectively.

The global sensitivity analysis of the relationship between the SEI and the input variables suggested that the final values of the index were positively correlated to the average forest cover of the watershed ($r = 0.51$), the water yield ($r = 0.34$), and the population served ($r = 0.25$), while being negatively correlated to the average value of the FWI ($r = -0.12$) (Table 2). One can also note the inverse relationship ($r = -0.64$) between fire danger and forest cover, which can be explained by opposing spatial patterns between those variables: For the 11 watersheds displaying fire danger values >35, the average forest cover was ~11%, which explains the limited influence on final exposure scores due to the highest FWI values observed in the least-forested watersheds. The spatial pattern of the SEI revealed a local effect of the number of consumers (Figure 6). Setting the forest cover input variable to its mean allows checking for the dependence of other layers, which revealed that the population was, indeed, the most important predictor of exposure, with a positive correlation ≈0.5 and higher index values spatially related to areas of greater population concentration, particularly in the southern part of Alberta (Figure 4b).

**Table 2.** Correlation coefficients between input variables and the exposure index. Values relative to the total amount of downstream consumers (TC), a raster grid of population distribution, were added to the analysis.

| Layer | Consumers | Water Yield | Forest Cover | Fire Hazard | SEI |
|---|---|---|---|---|---|
| **Consumers (TC)** | 1 | | | | |
| **Water yield (Q)** | 0.09 | 1 | | | |
| **Forest cover (FOR)** | −0.19 | 0.32 | 1 | | |
| **Fire danger (FID)** | −0.05 | −0.63 | −0.64 | 1 | |
| **Exposure index (SEI)** | 0.25 | 0.34 | 0.51 | −0.12 | 1 |

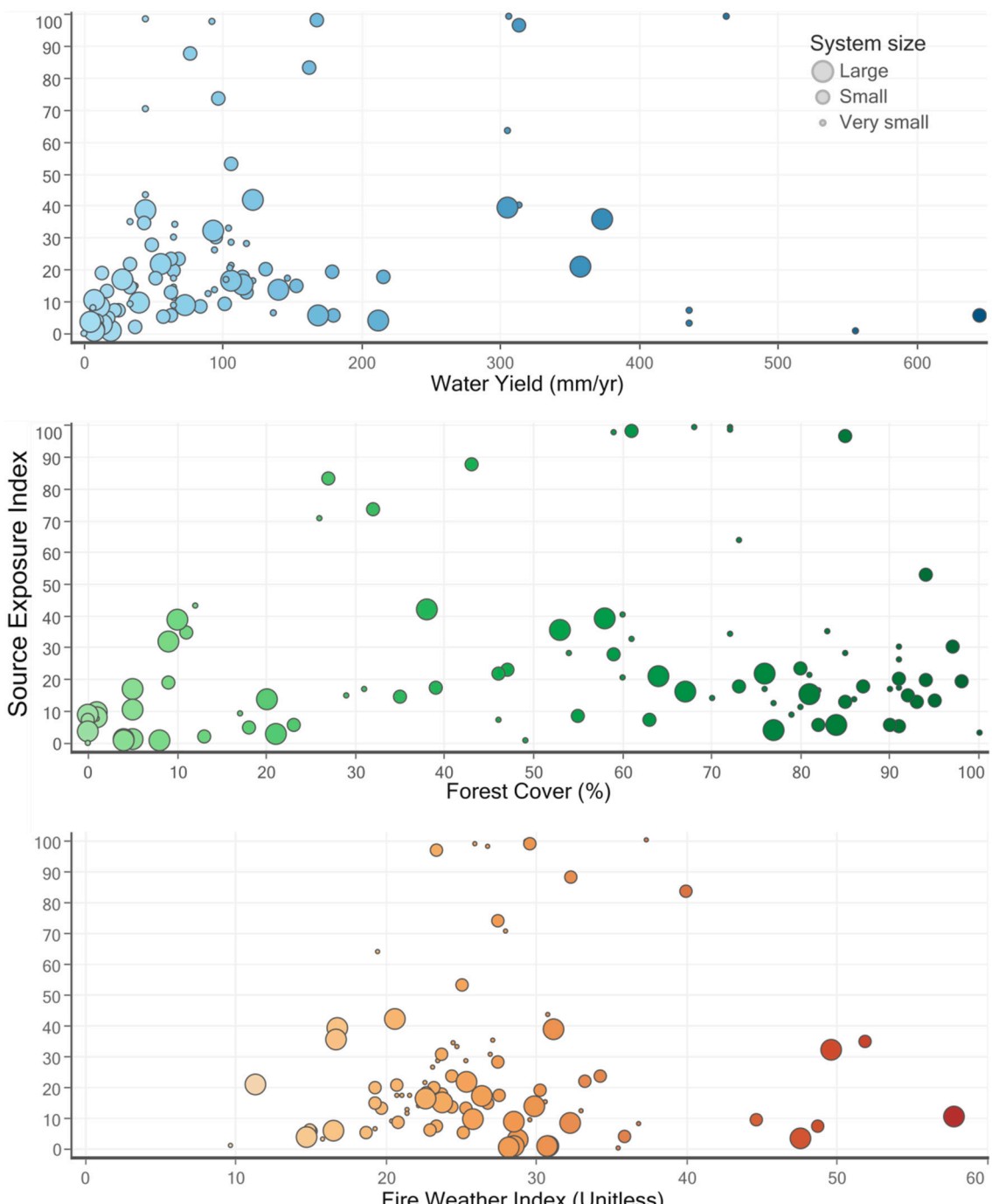

**Figure 6.** Scatterplots showing the relationship between the SEI score per source watershed and the three variables used to compute the index, namely the water yield, the forest cover, and the Fire Weather Index. The size of the dots are scaled by number of consumers served per drinking-water treatment plant, with very small, small, and large systems serving less than 500, 500 to 5000, and more than 5000 people, respectively. The shade of the dots helps to visualize change in the values of the different variables, darker tones showing greater values than lighter tones.

## 4. Discussion

The Source Exposure Index was conceived as an adaptable and generic tool for the evaluation of drinking-water supply exposure from wildfires. Our model relies on a minimum amount of data related to the location of drinking-water utilities and their consumers, water yield per unit area, percent

forest cover, and fire danger. Our approach created an exposure index based on the combination of these variables within each forested watershed supplying drinking water in Alberta. Our results show that drinking water source watersheds located in the southern region (i.e., Rocky Mountains) of the province had higher exposure scores, that the forest cover and the total number of downstream consumers drive the exposure scores, and that the source watersheds of smaller drinking-water systems are more exposed comparatively to larger systems.

The creation of our index followed the same general logic as F2F, although fully reproducing this latter method would have been difficult due to major differences in data availability and structure. However, given the potential consequences that wildfires can have on drinking-water sources, it is crucial to obtain such first-order estimates provided by the *SEI*, even if not all of the data is available. This situation led to several differences between the two indices. First, the exposure model for Alberta focuses on threats from wildfire only. Using the FWI as a fire hazard proxy instead of a probability layer comparable to the US Wildland Fire Potential is the most important difference. The FWI System provides a suite of meaningful and easy-to-compute indicators of fire hazard. Hydrologic information was also treated differently; F2F used what was essentially the result of a water-budget model (i.e., precipitation-evapotranspiration) [71], whereas we used actual water yield information collected from water monitoring gauges. Comparatively, our approach was based on historical measurements of streamflow, which offer the potential to assess alternate scenarios. For example, once compiled, we could use the base data to assess how the regional pattern of exposure changes during dry years with exceptionally low annual flows.

## 4.1. Wildfires and Drinking-Water Security in Alberta

For the past 10 years, the province of Alberta has experienced ~1500 wildfires per year for an average ~280,000 hectares in area burned each year [72]. However, there was large variability in their effects, and while most fires remained of limited concern, some of these were particularly impactful from a socio-economic perspective. For instance, the Flat Top Complex of 2011, though not particularly large by Canadian standard, exhibited unpredictable behavior and destroyed more than 400 structures in the Town of Slave Lake. Similarly, the Horse River fire of 2016 burned almost 600,000 hectares near the city of Fort McMurray, destroying or damaging nearly 2000 structures and forcing the evacuation of 88,000 people, for an estimated ~$9-billion of insurable losses [73]. This fire has resulted in significant municipal water-treatment costs, including a pre-cautionary three-month boil water order and increased annual treatment chemical consumption of approximately 50% [74,75], thereby illustrating existing wildfire risks to water supply in the province.

The province of Alberta relies on a number of environmental regulations that have put the protection of drinking water source watersheds at the forefront of environmental management, including: Alberta Environment's Drinking Water Program [76], the Alberta Water Act [77], Water for Life [78], and the Standards and Guidelines for Municipal Waterworks [79]. These policies and regulatory frameworks aim at public health protection through the provision of safe drinking water. The province of Alberta also relies on a number of land and wildfire management practices and forest conservation policies, such as the Forest and Prairie Protection Act [80] and the Land Use Framework [81] to protect communities and other values at risk from wildfires. The protection of source watersheds is the third priority of the Alberta Fire Management Branch [82]. The environmental regulatory context of Alberta therefore lends support to the use of the *SEI* as a tool to identify those source watersheds with higher exposure to wildfires and where source water protection (SWP) planning should account for post-fire water contamination.

SWP planning involves the identification of threats to source-water supplies, the evaluation of risks, and the implementation of management actions to ensure that the risks to water quality and quantity are prevented or minimized [83]. SWP plan development can be challenging. Delays in SWP planning—especially in small systems (i.e., those serving 5000 consumers or less)—have been attributed to the inherent technical complexity of such endeavors, challenges in engaging all relevant

stakeholders, and the costs associated with evaluating and implementing mitigation measures [84–86]. The relatively scant analyses of cost-benefit suggest the potential to avoid tens to hundreds of millions of dollars in costs through watershed protection against detrimental fire activity [87,88]. It should be noted, however, that many of those estimates pertain to complete avoidance, rather than mitigation of the impacts associated with severe wildfire, which is likely a more reasonable target for SWP-focused actions. Notably, the *SEI* can be used as a decision-support tool for informing such decisions through identification of watersheds and community water supplies with the greatest wildfire exposure.

### 4.2. Accounting for Wildfires in the Future of Canada's Water Security

Wildfire is increasingly cited as an emerging source of risks to water security [89,90], including in the Canadian water sector where future investments in wildfire risk resilience will be paramount [91]. Efforts to develop a Canadian water security assessment framework, with a strong focus on drinking-water safety will provide a national platform that heavily relies on the use of spatial environmental indicators, such as the *SEI* [92,93]. The framework outlined herein could be applied at a national scale to complement existing national water indicators [94,95] to specifically identify community watersheds with the greatest exposure to wildfires. Other aspects of freshwater supply can be addressed using the *SEI* with minor adaptation; for instance, the location of waterworks' intakes can be substituted with any value at risk, such as a reservoir, a lake used for recreation, or an endangered riverine ecosystem.

At least four of the forested watersheds in our data are a source of drinking water for First Nations, Inuit, or Metis communities, with a SEI score ranging from 5 to 50. Chronic water insecurity of First Nations, Inuit, or Metis communities of Canada, illustrated by the plethora of long-term boil-water advisories, has been a source of tension for decades [96,97]. These difficulties to access clean drinking-water expose Indigenous people to greater water-related health issues, on top of management difficulties inherent to small distribution systems, as noted earlier [98]. Many Indigenous communities are located in fire-prone forests [99], and though wildfire risks faced by those communities have been increasingly addressed [100], there is to date no formal assessment of the additional threat that wildfires would pose to existing water issues. Beyond potential drinking-water issues, it is noteworthy that those communities also depend on water resources for food (e.g., fish) and other uses (e.g., transportation, spiritual value). This creates additional concerns, as several studies have shown increases in heavy metals following wildfires, occasionally rising to dangerous levels for human health [101–104]. The *SEI* provides an easy-to-deploy tool to rapidly identify key regions where the range of effects (e.g., water supply, water quantity, and aquatic ecology) from wildfires may threaten the ensemble of freshwater ecosystem services that First Nations, Inuit, and Metis communities depend upon.

Ongoing global change will likely put Canadian forests under higher pressure from human activities and climatic variations, with wildfire activity and hydrologic extremes likely to increase in frequency and severity [49,105–107]. Projected changes in precipitation patterns, water availability and quality, and pressures on freshwater resources will also likely increase with Canada-wide human expansion (e.g., urban sprawl, population growth, natural resource exploitation) and its need for larger water amounts [27,91,108]. This combination will likely expose an increasing number of source watersheds to wildfire hazard, which in turn may lead to increased treatment challenges and upsets, and even boil-water advisories, service disruptions, or outages [109,110]. The flexibility of our index can be advantageously used to combine the *SEI* with the increasing availability of regional environmental change projection data (e.g., FWI forecasts) and water security indicators (e.g., Risk-Based Basin Analysis) so that future land and water governance policies in Canada can better address the effects of wildfires on particularly endangered source watersheds [93,95].

### 4.3. Limitation and Improvements

Our approach, though informative, does have some limitations. Firstly, solely using a distance variable as a proxy to account for downstream dilution processes implies a linear and unique watershed

response to wildfire [111]. It has been shown that the nature and the spatial arrangement of hydrologic features in a watershed partially control its capacity to buffer, or on the contrary, to trigger or accelerate post-fire hydrologic response. Integrating the diversity, the density, and the connectivity of existing hydrologic features could help better represent the exposure of a watershed to wildfire [112,113].

Secondly, the FWI System is a weather-based index that does not explicitly incorporate information on flammable vegetation (i.e., fuels), ignitions, and topography, which precludes the computation of specific measures of fire behavior influencing post-fire hydrologic response (i.e., size and severity). Combining FWI information with spatial data from the Canadian Fire Behavior Prediction System could help refine exposure scores, but reliable fuels data required for FBP System computation is not always available or up to date. This system indeed integrates forest fuel information and provides metrics according to potential fire intensity that could be used as a proxy to fire severity, either as depth of burn or biomass consumed, which is a critical factor of post-fire hydrologic response [114].

Finally, our index only characterizes the exposure of source watersheds, and does not provide any insight regarding the impacts to water quality and treatability, or the vulnerability of specific downstream drinking-water systems. Although Canadian (and North American) drinking-water systems are generally equipped to face some of the source water challenges associated with severe wildfires, the need for additional treatment infrastructure or elevated operational costs post-fire will have to be eventually factored into a larger risk assessment tool [16,115].

## 5. Conclusions

We have presented a large-scale analysis of drinking-water source exposure to wildfire hazard in Alberta, Canada. Our approach adapted a US framework and made it more generic, thus more broadly applicable to other regions where hydrological resources are exposed to fire activity. Our results show that the forest cover associated with the distribution of water consumers drove exposure levels, making North-central Alberta and the Southern Rocky Mountains particularly exposed. Those results are logical and well-illustrated by the consequences of recent extreme wildfires that occurred in the province, such as the Horse River fire in Fort McMurray, whose impact on water supply was extensively reported in the media [74,116]. They are also exemplified by the recent publication of source-water protection plans integrating wildfire threats to the drinking-water supply for the largest cities in Alberta [117,118]. However, not only are environmental conditions worsening due to global change, but the exposure of communities is increasing by virtue of increasing consumer demand. Our exposure model represents a customizable basis for a comprehensive national risk analysis of community water systems to wildfires. With the foreseen increase in the number of wildfires that could cause negative human impacts, such a simple tool can help quickly project environmental conditions using "what-if" scenarios, thereby facilitating the identification of watersheds at risk, leading to the design of tailored and cost-effective resilience strategies.

**Author Contributions:** Conceptualization, U.S., M.B.E., M.D.F., D.P.D., and M.-A.P.; methodology, F.-N.R., K.D.B., U.S., D.P.D., and M.B.E.; formal analysis, F.-N.R., K.D.B., M.-A.P., U.S., and M.B.E.; writing—original draft preparation, All Authors; writing—review and editing, All Authors; visualization, F.-N.R.

**Funding:** The preliminary work for this study was funded by the Canadian Water Network, Alberta Environment and Sustainable Resources Development, and Alberta Environment and Parks through the project "Management of Wildfire Risk to Municipal Waterworks Systems in Alberta". Further financial support for the publication of this work was provided by the Canadian Partnership for Wildland Fire Science as part of the Global Water Futures research initiative, led by the National Hydrology Research Centre at the University of Saskatchewan and funded by the Canada First Research Excellence Fund.

**Acknowledgments:** The authors want to thank Jourdan Bird who assisted in the collection of drinking-water treatment plant data.

**Conflicts of Interest:** The authors declare no conflict of interest.

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
