# Peer review of "A Regional-Scale Index for Assessing the Exposure of Drinking-Water Sources to Wildfires"

_forests, doi:10.3390/f10050384_

Round 1

Reviewer 1 Report

The Authors present a spatial index of watershed exposure to wildfires in the province of Alberta, utilizing a multi-criteria analysis design that integrates provincial forest cover, fire danger, source water volume, source water origin, and population served—which the Authors dub “the Source Exposure Index (SEI).” The Authors aim to address the lack of adequate tools for assessing potential wildfire-associated risks to municipal drinking-water-systems served by forested catchments outside the USA by building off of methods used in 2009 in the eastern USA and subsequently modified in the USFS Forests to Faucets (F2F) initiative. The authors indicated that the utility of the F2F is constrained by its US-centric design and differences in input data availability, and assert that SEI differs from F2F and other efforts by utilizing different source data, integrating source data differently, and identifying the spatial location of drinking-water-supply intakes and relating these spatially to forested water sources and their wildfire danger history. Overall, the manuscript is well-written and a timely discussion of a salient topic. Conceptually, the paper is sound and I particularly enjoyed the discussion regarding the potential utility of the SEI for mitigating impacts to First Nations' populations. The Authors’ use of figures and tables greatly adds to the clarity of the paper. A major weakness of this manuscript is the discussion of the forest cover and fire danger inputs in the methods section. Several concepts need to be clarified in these paragraphs and some limitations with these layers need to be acknowledged in the methods section, in addition to the discussion of improving the fire danger measure in the discussion section. In particular, clarification of what “gaps” are and how they were dealt with (i.e. interpolation) needs more discussion.

Broad comments:

The discussion of the forest cover layer (Section 2.2.3) is a bit confusing and requires some additional detail.

(1)   While it is obvious in the reference section that the authors are using the 2010 version, it might be useful to have the version year mentioned in-text to enhance the clarity for the reader.

(2)   Between discussion of the resolution and the classes kept, it might be helpful to mention that there are 11 classes across the province of Alberta to give context about the options.

(3)   A limitation of the dataset that warrants acknowledgement here is the accuracy assessment for the classes used in this study. I would recommend at least discussing the low accuracy (30%) of the shrub class. How did the authors deal with “many shrub polygons are actually forest (beyond the new cutblocks and brunt areas), especially in the North” (ABMI 2010, p. 2)? What kind of impact does this inaccuracy have on your forest-focused study, if forests were typically mis-labeled as shrubland? Perhaps you can address this and provide a justification that ABMI is still useful for regional assessments, as was described in ABMI 2010 (top of p.3). Did most/some of these areas with low accuracy ended up being outside the WOIs, potentially even because of this classification issue?

(4)   Perhaps my strongest recommendation for strengthening the paper is clarifying the discussion of how large gaps were dealt with and the interpolation process (Lines 207-209). It appears that forest areas harvested or burned between 2000 and 2010 were assigned to the “Shrub” LC class and designated as “cutblocks” or “burnt areas” (ABMI 2010, p. 2). Are these the “gaps” referenced, or are “gaps” a product of incorporating the National Fire Database polygon layer and overwriting the default “shrub” designation? Are the “gaps” some other areas altogether that are missing vegetation cover data?

(5)   I would also request some more detail describing the interpolation process. I am familiar with interpolation being used to fill in gaps between point data, yet it appears that it is being used to fill in “missing data” between polygons. Is the interpolation process supposed to fill in what used to be forested areas, or is it meant to represent the current vegetation post-fire (e.g., capture succession), and were time-since-fire or climate conditions considered? Are nearest neighboring polygons being used to fill in a 30-m area, or the entire fire perimeter polygon? Providing a hypothetical example might help clarify what was done. At present it is very unclear a) what exactly constitutes a “gap”, b) why these gaps needed to be “filled in”, and c) what actual process was used to fill them. Since this layer is key to calculating the percent of a WOI that is forested, it is important that the reader be able to understand how this layer was developed (particularly those modifications made by the authors and therefore not documented in the ABMI documentation) and its limitations.

Section 2.2.4. Fire danger.

I found there to be a lot of detailed information in this section regarding the different components, but that the logic/ties to the objectives was somewhat unclear overall. 

·        While the FWI index appears to be an appropriate proxy for fire danger, the reason for utilizing the 95% percentile, and how it exactly relates to “wildfire exposure”/objective (a), warrants some discussion. Why was a metric like “days/portion of fire season above X index” not used?

·        For those unfamiliar with the ‘cffdrs’ R package, a short description of what that package did and how it creates the layer would help clarify how and why that package was used.

Specifics/line-by-line edits:

Line 143-144: The statement that “Historically, large fires occurred more frequently in the norther boreal region;…” needs a reference. Fire size and frequency get conflated in Figure 1.

Line 144-145: “…; however, fire also remains the dominant natural disturbance agent in the southwestern headwaters of the Rocky Mountain forest region (Figure 1)” also requires evidence to back the claim.

Line 179-180: Table 1: The centering of the information in each cell in the table makes it somewhat challenging to read, particularly the last three variables. Perhaps align the variable name with the top of the row so it’s easier to discern where the proxy and source information begin and end for a given variable?

Figure 5. Line 324-5: The “System Size” legend appears to be outside of the image in my version. Lowering the legend will help with the cleanliness of the graph.

Lines 371-3: Something appears to be slightly off with sentence structure, verb tense, or article use. Please revise for clarity.

Line 383: This sentence is somewhat unclear. Perhaps revise to: “The protection of source watersheds is the third priorit[y] [of] the Alberta Fire Management Branch.”

 Lines 392-395: Consider revising sentence to make it more active and clarify the idea for the reader. Potentially reorder subjects and verbs such that “…suggest the potential [to avoid] tens to hundreds of millions of dollars [in costs] through watershed protection against…”

Line 421-423: For “Several studies conducted in Canada…dangerous levels for human health [93].” please include more references if possible.

Line 424-425: Please revise this sentence to clarify the idea, particularly focusing in on “…the cumulative impact wildfires can threaten…”

Line 429-430: the use of “national-scale anthropogenic expansion” is very broad and it is  difficult to understand what the sentence is implying, especially since anthropogenic typically refers to environmental pollution or climate change. Is the intent to discuss residential expansion, population growth, the expansion of human influence (e.g., anthropogenic climate change, emissions) on landscapes in Canada?

Line 444: Please revise “could help better representing the exposure of a…” to “could help better represent the exposure of a…”

Line 466-467: The assertion that “Those results are logical and well-illustrated by the consequences of recent extreme wildfire events that happened in the province.” requires evidence, either in the form of referencing post-fire reports or mentioning some examples of recent impactful wildfires.

Line 473: I’d recommend that the Authors considered whether to use “wildfire disaster” or “wildfire events” in this discussion point (i.e. “With the foreseen increase in the number of wildfire disasters…”). To me, “disaster” implies that noted increases in conditions that promote wildfire ignitions and spread (e.g., exposure to a hazard) will result in the outcome of the social or biophysical system being overwhelmed (e.g., high social or biophysical vulnerability resulting in the loss of life, property, ecosystem services/function, etc.) resulting in a disaster. This is a potentially more robust claim than discussing increasing trends in exposure to and experiences with wildfire events.

References:

There are several formatting errors in the reference section that should be revised to enable readers to find sources/data.

Most importantly, placing appropriate punctuation between Author(s) and Title would help readers track down data sources and referenced material.

Other typical errors include capitalization of words in paper Titles that are not proper nouns or do not follow a colon (e.g. Line 496-497: “State of the world’s freshwater ecosystems: Physical, chemical, and biological changes.”), Author names (e.g., Foley, J. a. àFoley, J. A.; “Robinne, F.-N.” to “Robinne, F. N.” [?]), book title capitalization (e.g., MHW’s Water Treatment; Principles and Design); and occasionally needing to capitalize the “a” in “Alberta” in Titles or Publisher Locations.

Author Response

Reviewer #1

Comments and Suggestions for Authors

The Authors present a spatial index of watershed exposure to wildfires in the province of Alberta, utilizing a multi-criteria analysis design that integrates provincial forest cover, fire danger, source water volume, source water origin, and population served—which the Authors dub “the Source Exposure Index (SEI).” The Authors aim to address the lack of adequate tools for assessing potential wildfire-associated risks to municipal drinking-water-systems served by forested catchments outside the USA by building off of methods used in 2009 in the eastern USA and subsequently modified in the USFS Forests to Faucets (F2F) initiative. The authors indicated that the utility of the F2F is constrained by its US-centric design and differences in input data availability, and assert that SEI differs from F2F and other efforts by utilizing different source data, integrating source data differently, and identifying the spatial location of drinking-water-supply intakes and relating these spatially to forested water sources and their wildfire danger history. Overall, the manuscript is well-written and a timely discussion of a salient topic. Conceptually, the paper is sound and I particularly enjoyed the discussion regarding the potential utility of the SEI for mitigating impacts to First Nations' populations. The Authors’ use of figures and tables greatly adds to the clarity of the paper. A major weakness of this manuscript is the discussion of the forest cover and fire danger inputs in the methods section. Several concepts need to be clarified in these paragraphs and some limitations with these layers need to be acknowledged in the methods section, in addition to the discussion of improving the fire danger measure in the discussion section. In particular, clarification of what “gaps” are and how they were dealt with (i.e. interpolation) needs more discussion.

We thank the reviewer for the positive feedback and the detailed review that was provided. We hope that the reviewer’s concerns are properly addressed hereafter.

Broad comments:

The discussion of the forest cover layer (Section 2.2.3) is a bit confusing and requires some additional detail.

We agree with the reviewer and we improved the section so what was done is made clearer to the reader.

#1: While it is obvious in the reference section that the authors are using the 2010 version, it might be useful to have the version year mentioned in-text to enhance the clarity for the reader.

Done, L.206

#2:  Between discussion of the resolution and the classes kept, it might be helpful to mention that there are 11 classes across the province of Alberta to give context about the options.

Done, L.207-209: “Based on the 11 land cover classes available in the dataset,”

#3: A limitation of the dataset that warrants acknowledgement here is the accuracy assessment for the classes used in this study. I would recommend at least discussing the low accuracy (30%) of the shrub class. How did the authors deal with “many shrub polygons are actually forest (beyond the new cutblocks and brunt areas), especially in the North” (ABMI 2010, p. 2)? What kind of impact does this inaccuracy have on your forest-focused study, if forests were typically mis-labeled as shrubland? Perhaps you can address this and provide a justification that ABMI is still useful for regional assessments, as was described in ABMI 2010 (top of p.3). Did most/some of these areas with low accuracy ended up being outside the WOIs, potentially even because of this classification issue?

The shrubland cover class was assimilated to the general forest cover layer presented in Figure 1d. We understand that the original paragraph is misleading: we selected the different forest types and shrubland cover polygons from the ABMI data and dissolved these into one general forest cover layer. In other words, shrublands were assimilated to forests in the forest cover layer that was created. Therefore, the issue underlined by the reviewer related to the misclassification of forest into shrubland was addressed by default while creating the layer.

As suggested by the reviewer, we mentioned that the ABMI product is useful for regional assessments, L.207: “…satellite imagery and developed for regional-scale environmental assessments [61].”

#4: Perhaps my strongest recommendation for strengthening the paper is clarifying the discussion of how large gaps were dealt with and the interpolation process (Lines 207-209). It appears that forest areas harvested or burned between 2000 and 2010 were assigned to the “Shrub” LC class and designated as “cutblocks” or “burnt areas” (ABMI 2010, p. 2). Are these the “gaps” referenced, or are “gaps” a product of incorporating the National Fire Database polygon layer and overwriting the default “shrub” designation? Are the “gaps” some other areas altogether that are missing vegetation cover data?

We agree that the explanation can be improved. We realize that we misused the word “interpolation”, as no interpolation process was used. What we refer to as “gaps” is a byproduct of creating our forest layer and those gaps represent non-forested areas according to the ABMI land cover classification.

However, it is common in many land cover products to find burnt areas classified as non-forested, especially in recently burnt regions. In the boreal region where fires have such a large influence on the landscape, this fact can bias forest-cover calculation and artificially reduce the total forest cover, as those burnt areas were forested and will eventually return to a forested stage. Simply put, it is a way to represent what would be the forest cover without recent fire history.

In order to create this “idealized” forest cover accounting for burnt areas, we merged the forest cover layer derived from ABMI with the information of the National Large Fire Database, thereby filling up the potential gaps (i.e., the difference between the two layers) that might have been erroneously classified as non-vegetation.

We updated the paragraph to make our explanation clearer, L.205-217: “Forest cover information was extracted from the Alberta Biodiversity Monitoring Institute (ABMI) Wall-to-wall 2010 land cover vector map,  which was derived from Landsat 30-meter satellite imagery and developed for regional-scale environmental assessments [59]. Based on the 11 land-cover classes available in the dataset, we selected the classes providing information on coniferous, broadleaf, and mixed forest types. We also included the shrubland cover class, as it often represents young or low-canopy forests and post-fire forest regeneration [61]. The polygons pertaining to those classes were then dissolved into a single-value forest cover layer and converted to a raster grid. We compared the ABMI land cover layer with large fire (≥ 200 ha) perimeters from the National Fire Database polygon layer between 1980–2016 [49], as it is common in many land-cover products to find burnt areas classified as non-forested, especially in recently burnt regions. Gaps (i.e., no data) in the forest cover where large fire perimeters overlapped were coded as forest as a simple way to account for post-fire vegetation recovery. We used this forest cover layer to calculate percent forest cover for each WOI (Figure 3b).

#5:  I would also request some more detail describing the interpolation process. I am familiar with interpolation being used to fill in gaps between point data, yet it appears that it is being used to fill in “missing data” between polygons. Is the interpolation process supposed to fill in what used to be forested areas, or is it meant to represent the current vegetation post-fire (e.g., capture succession), and were time-since-fire or climate conditions considered? Are nearest neighboring polygons being used to fill in a 30-m area, or the entire fire perimeter polygon? Providing a hypothetical example might help clarify what was done. At present it is very unclear a) what exactly constitutes a “gap”, b) why these gaps needed to be “filled in”, and c) what actual process was used to fill them. Since this layer is key to calculating the percent of a WOI that is forested, it is important that the reader be able to understand how this layer was developed (particularly those modifications made by the authors and therefore not documented in the ABMI documentation) and its limitations.

See the answer to the two previous comments. We misused the word “interpolation”, and we didn’t try to capture any succession dynamic.

Section 2.2.4. Fire danger.

I found there to be a lot of detailed information in this section regarding the different components, but that the logic/ties to the objectives was somewhat unclear overall. 

#6: While the FWI index appears to be an appropriate proxy for fire danger, the reason for utilizing the 95% percentile, and how it exactly relates to “wildfire exposure”/objective (a), warrants some discussion. Why was a metric like “days/portion of fire season above X index” not used?

The reasoning behind the use of the 95th percentile is to capture the relative frequency of days conducive to high or extreme wildfire behavior. Wildfires are very much driven by extreme conditions and it is during those days of particularly hot, dry, and windy conditions (captured by the FWI) that most of the area burns in the boreal forest.

The use of the 95th percentile was informed by previous wildfire-climate studies in boreal Canada (Parisien et al. 2014; Wang et al. 2015). These studies showed that, with all other factors being equal, the spatial and temporal distribution of 90th or 95th percentile weather conditions better correlate to fire activity than mean or median conditions. Also, because percentiles in this study are calculated for the fire season (rather than all year), their values are equivalent to the ‘days/portion of the fire season above X’ that is mentioned by the reviewer.

Parisien, M.A., Parks, S.A., Krawchuk, M.A., Little, J.M., Flannigan, M.D., Gowman, L.M. and Moritz, M.A., 2014. An analysis of controls on fire activity in boreal Canada: comparing models built with different temporal resolutions. Ecological Applications, 24(6), pp.1341-1356.

Wang, X., Thompson, D.K., Marshall, G.A., Tymstra, C., Carr, R. and Flannigan, M.D., 2015. Increasing frequency of extreme fire weather in Canada with climate change. Climatic Change, 130(4), pp.573-586.

 #7: For those unfamiliar with the ‘cffdrs’ R package, a short description of what that package did and how it creates the layer would help clarify how and why that package was used.

We agree with the reviewer. We updated the sentence as follow L.234-236: “The gridded FWI System components were calculated from the gridded weather data using the fwiRaster function from the ‘cffdrs’ R package [62], which was developed to calculate the components of the Canadian Forest Fire Danger Rating System [61]. ”

Specifics/line-by-line edits:

#8: Line 143-144: The statement that “Historically, large fires occurred more frequently in the norther boreal region;…” needs a reference.

We revised the sentence to improve clarity of the idea. It now reads L.146-148: “Historically, large fires occurred more frequently in the northern boreal region relative to the southern areas of the province.”

We also added the following references to support this idea, as suggested: Heon et al., PNAS, 2014; Tymstra et al., Int. J. Wildland Fire, 2007.

#9: Fire size and frequency get conflated in Figure 1.

We updated the figure so fire size and frequency are easier to discern.

#10: Line 144-145: “…; however, fire also remains the dominant natural disturbance agent in the southwestern headwaters of the Rocky Mountain forest region (Figure 1)” also requires evidence to back the claim.

We added the following reference to support this idea, as suggested: Rogeau, M.-P.; Flannigan, M.D.; Hawkes, B.C.; Parisien, M.-A.; Arthur, R. Spatial and temporal variations of fire regimes in the Canadian Rocky Mountains and Foothills of Southern Alberta.Int. J. Wildland Fire 2016,25, 1117.

#11: Line 179-180: Table 1: The centering of the information in each cell in the table makes it somewhat challenging to read, particularly the last three variables. Perhaps align the variable name with the top of the row so it’s easier to discern where the proxy and source information begin and end for a given variable?

The reviewer is right; the original table is difficult to read. As suggested by another reviewer, we added lines to separate the rows and we added space between items.

#12: Figure 5. Line 324-5: The “System Size” legend appears to be outside of the image in my version. Lowering the legend will help with the cleanliness of the graph.

Done

#13: Lines 371-3: Something appears to be slightly off with sentence structure, verb tense, or article use. Please revise for clarity.

We revised the sentence to improve clarity, as suggested, L387-390: “Comparatively, our approach was based on historical measurements of streamflow, which offer the potential to assess alternate scenarios. For example, once compiled, we could use the base data to assess how the regional pattern of exposure changes during dry years with exceptionally low annual flows.”

#14: Line 383: This sentence is somewhat unclear. Perhaps revise to: “The protection of source watersheds is the third priorit[y] [of] the Alberta Fire Management Branch.”

Fixed as the reviewer suggested

#15: Lines 392-395: Consider revising sentence to make it more active and clarify the idea for the reader. Potentially reorder subjects and verbs such that “…suggest the potential [to avoid] tens to hundreds of millions of dollars [in costs] through watershed protection against…”

Fixed as the reviewer suggested

#16: Line 421-423: For “Several studies conducted in Canada…dangerous levels for human health [93].” please include more references if possible.

We added three additional references (Ranalli et al. 2002; Silins et al., 2009; and Biswas et al., 2007) to support this idea, as suggested.

#17: Line 424-425: Please revise this sentence to clarify the idea, particularly focusing in on “…the cumulative impact wildfires can threaten…”

We revised the sentence to improve clarity, as suggested, L451-454: “The SEI provides an easy-to-deploy tool to rapidly identify key regions where the range of effects (e.g., water supply, water quantity, aquatic ecology) from wildfires may threaten the ensemble of freshwater ecosystem services that First Nations, Inuit, and Metis communities depend upon.”

#18: Line 429-430: the use of “national-scale anthropogenic expansion” is very broad and it is  difficult to understand what the sentence is implying, especially since anthropogenic typically refers to environmental pollution or climate change. Is the intent to discuss residential expansion, population growth, the expansion of human influence (e.g., anthropogenic climate change, emissions) on landscapes in Canada?

The reviewer is right; our intent is to discuss the increasing human footprint on Canadian landscapes. We revised this sentence accordingly, L.458-460: “…with Canada-wide human expansion (e.g. urban sprawl, population growth, natural resource exploitation) and…”

#19: Line 444: Please revise “could help better representing the exposure of a…” to “could help better represent the exposure of a…”

Done

#20: Line 466-467: The assertion that “Those results are logical and well-illustrated by the consequences of recent extreme wildfire events that happened in the province.” requires evidence, either in the form of referencing post-fire reports or mentioning some examples of recent impactful wildfires.

We updated the discussion and the conclusion so those refer to recent impactful wildfires, as follow, L.392-402: “For the past 10 years, the province of Alberta has experienced 1500 wildfires per year for an average 280,000 hectares in area burned each year [70]. However, there was large variability in their effects, and while most fires remained of limited concern, some of these have been particularly impactful from a socio-economic perspective. For instance, the Flat Top Complex of 2011, though not particularly large by Canadian standard, exhibited unpredictable behavior and destroyed more than 400 structures in the Town of Slave Lake. Similarly, the Horse River fire of 2016 burned almost 600,000 hectares near the city of Fort McMurray, destroying or damaging nearly 2,000 structures and forcing the evacuation of 88,000 people, for an estimated ~$9 billion of insurable losses [71]. This fire has resulted in significant municipal water-treatment costs, including a pre-cautionary three-month boil water order and increased annual treatment chemical consumption of approximately 50% [72,73], thereby illustrating existing wildfire risks to water supply in the province.”

and L.496-498: “Those results are logical and well-illustrated by the consequences of recent extreme wildfires that occurred in the province, such as the Horse River fire in Fort McMurray whose impact on water supply was extensively shared in the media [74,116]”.

#21: Line 473: I’d recommend that the Authors considered whether to use “wildfire disaster” or “wildfire events” in this discussion point (i.e. “With the foreseen increase in the number of wildfire disasters…”). To me, “disaster” implies that noted increases in conditions that promote wildfire ignitions and spread (e.g., exposure to a hazard) will result in the outcome of the social or biophysical system being overwhelmed (e.g., high social or biophysical vulnerability resulting in the loss of life, property, ecosystem services/function, etc.) resulting in a disaster. This is a potentially more robust claim than discussing increasing trends in exposure to and experiences with wildfire events.

We agree with the reviewer and have opted to change “wildfire disaster” to simply “wildfires”. The sentence to which the reviewer refers to was also rephrased as follows, L503-504: “With the foreseen increase in the number of wildfires that could cause negative human impacts”.

 References:

#22: There are several formatting errors in the reference section that should be revised to enable readers to find sources/data.

Most importantly, placing appropriate punctuation between Author(s) and Title would help readers track down data sources and referenced material.

Other typical errors include capitalization of words in paper Titles that are not proper nouns or do not follow a colon (e.g. Line 496-497: “State of the world’s freshwater ecosystems: Physical, chemical, and biological changes.”), Author names (e.g., Foley, J. a. àFoley, J. A.; “Robinne, F.-N.” to “Robinne, F. N.” [?]), book title capitalization (e.g., MHW’s Water Treatment; Principles and Design); and occasionally needing to capitalize the “a” in “Alberta” in Titles or Publisher Locations

We reviewed the formatting of all our references and update it according to the reviewer’s comment. The hyphen was left for dual first names (e.g. François-Nicolas, Marc-André).

Reviewer 2 Report

Dear Authors,

Your paper is well written, clear and easy to understand.  It was a pleasure to read.  Thank you.

The Source Exposure Index provides a method for ranking relative risk that wildfire poses to drinking water supplies in Alberta, Canada.  A major improvement to the paper would be to provide estimates of actual risk.  Given this is difficult, perhaps include an example in the discussion of recent costs incurred by a water utility due to wildfire and some recent wildfire statistics for Alberta.

Minor edits:

Line 42 "The index can help regional authorities prioritizing the allocation of risk management resources to mitigate adverse impacts from wildfire."

change to

"The index can help regional authorities prioritize the allocation of risk management resources to mitigate adverse impacts from wildfire."

Line 124  Abbreviations should be spelled out the first time they are used. MASL - meters above sea level

Line 154 Extra space in (92%)

Table 1:  Additional space or row lines are needed to delineate rows in Table 1.  It is very difficult to read the Proxy and Source columns.

Line 198: TYPO should be Figure 2a, not 3a

Section 2.2.4 - I think the paper would be improved with a brief introduction to the Canadian Forest Fire Danger Rating System (CFFDRS).  I also recommend rewriting some of the sentences to avoid over use of "index", it is used 4 times in one sentence.

Figure 3. The dark blue and dark green colors used in this graphic make it difficult to read the text.  Perhaps employ more shapes to improve clarity.

Line 313 -   Sentence is not clear - not sure what the 77% is referring to. "These watersheds primarily (77%) serve small to very small communities across the north-central region of the province."    

Figure 6.  There is shade variation in the symbols, would be good to indicate the meaning in the text.

lines 370-373: The sentence is too long, also I think you need to change analyses to analyze.    

lines 443-445. "Integrating the diversity, the density, and the connectivity of existing hydrologic features could help better representing the exposure of a watershed to wildfire [101,102]"

sentence is missing the period change to:

lines 443-445. "Integrating the diversity, the density, and the connectivity of existing hydrologic features could help better represent the exposure of a watershed to wildfire [101,102]."

Author Response

Reviewer #2

Comments and Suggestions for Authors

Dear Authors,

Your paper is well written, clear and easy to understand.  It was a pleasure to read.  Thank you.

We thank the reviewer for this positive feedback and we appreciate the time taken to review this manuscript. We did our best to address the reviewer’s concerns below.

The Source Exposure Index provides a method for ranking relative risk that wildfire poses to drinking water supplies in Alberta, Canada.  A major improvement to the paper would be to provide estimates of actual risk. 

#1: Given this is difficult, perhaps include an example in the discussion of recent costs incurred by a water utility due to wildfire and some recent wildfire statistics for Alberta.

We updated the section 4.1 in the discussion to address this comment. It now reads as follow L.392-402: “For the past 10 years, the province of Alberta has experienced 1500 wildfires per year for a 280,000 hectares in area burned each year, on average [72]. However, there was large variability in their effects, and while most fires remained of limited concern, some of these have been particularly impactful from a socio-economic perspective. For instance, the Flat Top Complex of 2011, though not extremely large by Canadian standard, had an unpredictable behavior and destroyed more than 400 structures in the Town of Slave Lake. Similarly, the Horse River fire of 2016 burned almost 600,000 hectares near Fort McMurray, destroying or damaging nearly 2,000 structures and forcing the evacuation of 88,000 people, for an estimated cost of ~$9 billion [73]. This latter fire have since posed significant challenges to municipal water treatment operations, imposing a three-month boil water order and increasing annual treatment costs by up to one million dollars [74,75], thereby illustrating existing wildfire risks to water supply in the province.”

Minor edits:

#2: Line 42 "The index can help regional authorities prioritizing the allocation of risk management resources to mitigate adverse impacts from wildfire." change to "The index can help regional authorities prioritize the allocation of risk management resources to mitigate adverse impacts from wildfire."

Done

#3: Line 124  Abbreviations should be spelled out the first time they are used. MASL - meters above sea level

Fixed

#4: Line 154 Extra space in (92%)

We have followed the recommendations of ‘The International System of Units’ (SI) throughout the manuscript. In this case, SI indicates the following: “When (sic) used, it is necessary to put a space between the number and the symbol %”. There has been no change to the manuscript in this regard.

#5: Table 1:  Additional space or row lines are needed to delineate rows in Table 1.  It is very difficult to read the Proxy and Source columns.

Done

#6: Line 198: TYPO should be Figure 2a, not 3a

Fixed

#7: Section 2.2.4 - I think the paper would be improved with a brief introduction to the Canadian Forest Fire Danger Rating System (CFFDRS).  I also recommend rewriting some of the sentences to avoid over use of "index", it is used 4 times in one sentence.

We re-worked the first section of this paragraph to address the reviewer’s comment. It now reads as follows L219-236:

“The fire danger data used in this study consisted of raster grids of the Fire Weather Index (FWI) System components. This system is one of two major components of the Canadian Fire Danger Rating System (CFFDRS), the other being the Fire Behavior Prediction (FBP) System (not considered in this study). The FWI System’s components are calculated from daily weather conditions (temperature, relative humidity, wind speed, and 24-hour precipitation); these may in turn used in conjunction with data representing flammable vegetation (i.e., fuels) and topography by the FBP System to calculate quantitative measures of fire behavior (e.g., rate of spread, fire intensity). The FWI System is composed of three fuel moisture codes and three fire behavior indices [60]. The three codes, the Fine Fuel Moisture Code (FFMC), the Duff Moisture Code (DMC), and Drought Code (DC) represent the fuel moisture of surface, intermediate, and deep soil layers, respectively. The Initial Spread Index (ISI) is a wind-based indicator of fire danger, whereas the Buildup Index (BUI) is chiefly drought based. The Fire Weather Index (FWI) is an integrated indicator of overall fire danger computed from the ISI and BUI. The Canadian fire-weather database, an interpolated raster product of daily fire weather at a 3-km resolution, was provided by the Canadian Forest Service from historical data, based on surface (i.e., weather station) observations between April 1 and September 30 from 1981 to 2010 [61]. The gridded FWI System components were calculated from the gridded weather data using the fwiRaster function from the ‘cffdrs’ R package [62], which was developed to calculate the components of the Canadian Forest Fire Danger Rating System [61].  ”

#8: Figure 3. The dark blue and dark green colors used in this graphic make it difficult to read the text.  Perhaps employ more shapes to improve clarity.

We updated the color scheme so the text is easier to read.

#9: Line 313 -   Sentence is not clear - not sure what the 77% is referring to. "These watersheds primarily (77%) serve small to very small communities across the north-central region of the province."    

Fixed. We spelled out 77 %. L329-330: “Seventy-seven percent of these…”

#10: Figure 6.  There is shade variation in the symbols, would be good to indicate the meaning in the text.

We added L351-352: “The shade of the dots helps to visualize change in the values of the different variables, darker tones showing greater values than lighter ones.”

#11: lines 370-373: The sentence is too long, also I think you need to change analyses to analyze.    

We revised the sentence. It now reads L387-390: “Comparatively, our approach was based on historical measurements of streamflow, which offer the potential to assess alternate scenarios. For example, once compiled, we could use the base data to assess how the regional pattern of exposure changes during dry years with exceptionally low annual flows.” as suggested.

#12: lines 443-445. "Integrating the diversity, the density, and the connectivity of existing hydrologic features could help better representing the exposure of a watershed to wildfire [101,102]" sentence is missing the period change to:

lines 443-445. "Integrating the diversity, the density, and the connectivity of existing hydrologic features could help better represent the exposure of a watershed to wildfire [101,102]."

Done

Round 2

Reviewer 1 Report

I am very pleased with the Authors' changes to the manuscript, particularly with the clarification of the forest cover information. Their response and reasoning for utilizing the 95% percentile for their fire danger data in the construction of the WOIs was excellent. Including a 1-2 sentence explanation in-text would clarify the reasoning for readers less familiar with fire models. The ideas presented in their response may be particularly valuable here especially the ideas in this segment: "...the 95th percentile is to capture the relative frequency of days conducive to high or extreme wildfire behavior. Wildfires are very much driven by extreme conditions and it is during those days of particularly hot, dry, and windy conditions (captured by the FWI) that most of the area burns in the boreal forest. The use of the 95th percentile was informed by previous wildfire-climate studies in boreal Canada (Parisien et al. 2014; Wang et al. 2015)." 

Overall, the new version of the manuscript is a pleasant read and any prior points of confusion in the narrative have been addressed. The addition of the new first paragraph to section 4.1 also couples nicely with the subsequent discussion of environmental regulations, SWP planning, and the utility of SEI as a decision-support tool. Thank you.

Line edits [suggested changes in bold]: 

Lines 84-87: "For instance, treatment of lower-quality source water coming from burned areas may result in increased [what?], challenging [who?: municipalities, governments, the coagulant producers?] to meet chemical coagulant demand [16]..."

Line 206: "(a) evaluate the exposure of source water supplying communities downstream of wildfire risk, and...

Line 387: "...; these may in turn be used in conjunction with data..."

Author Response

Forests —Robinne et al. — Answers to reviewers’ comments

We address hereafter each individual comment. For clarity purpose, answers are written in blue and the in-text revisions of the manuscript are in “italic”.

--------------------------------------------------------------------------

Reviewer #1 – Round 2

Comments and Suggestions for Authors

I am very pleased with the Authors' changes to the manuscript, particularly with the clarification of the forest cover information.

We thank the reviewer for the positive feedback and we are glad our updates were satisfying.

Their response and reasoning for utilizing the 95% percentile for their fire danger data in the construction of the WOIs was excellent. Including a 1-2 sentence explanation in-text would clarify the reasoning for readers less familiar with fire models. The ideas presented in their response may be particularly valuable here especially the ideas in this segment: "...the 95th percentile is to capture the relative frequency of days conducive to high or extreme wildfire behavior. Wildfires are very much driven by extreme conditions and it is during those days of particularly hot, dry, and windy conditions (captured by the FWI) that most of the area burns in the boreal forest. The use of the 95th percentile was informed by previous wildfire-climate studies in boreal Canada (Parisien et al. 2014; Wang et al. 2015)."

We agree that this information is valuable; we therefore updated the section accordingly, L246-249: “We used the 95th percentile to capture the relative frequency of days conducive to high or extreme wildfire behavior. Wildfires are very much driven by extreme conditions and it is during those days of particularly hot, dry, and windy conditions (captured by the FWI) that most of the area burns in the boreal forest [68,69].”

Overall, the new version of the manuscript is a pleasant read and any prior points of confusion in the narrative have been addressed. The addition of the new first paragraph to section 4.1 also couples nicely with the subsequent discussion of environmental regulations, SWP planning, and the utility of SEI as a decision-support tool. Thank you.

Line edits [suggested changes in bold]:

Lines 84-87: "For instance, treatment of lower-quality source water coming from burned areas may result in increased [what?], challenging [who?: municipalities, governments, the coagulant producers?] to meet chemical coagulant demand [16]..."

Updated, L72-74: “For instance, treatment of lower-quality source water coming from burned areas may result in difficulties for treatment plants to meet chemical coagulant demand

Line 206: "(a) evaluate the exposure of source water supplying communities downstream of wildfire risk, and...

Changed L113

Line 387: "...; these may in turn be used in conjunction with data..."

Changed L224